

# miR2Trait: an integrated resource for investigating miRNA-disease associations

Poornima Babu and  Ashok Palaniappan

Department of Bioinformatics, School of Chemical and Biotechnology, SASTRA University, Thanjavur, Tamil Nadu, India

## ABSTRACT

MicroRNAs are key components of cellular regulatory networks, and breakdown in miRNA function causes cascading effects leading to pathophenotypes. A better understanding of the role of miRNAs in diseases is essential for human health. Here, we have devised a method for comprehensively mapping the associations between miRNAs and diseases by merging on a common key between two curated omics databases. The resulting bidirectional resource, miR2Trait, is more detailed than earlier catalogs, uncovers new relationships, and includes analytical utilities to interrogate and extract knowledge from these datasets. miR2Trait provides resources to compute the disease enrichment of a user-given set of miRNAs and analyze the miRNA profile of a specified diseasome. Reproducible examples demonstrating use-cases for each of these resource components are illustrated. Furthermore we used these tools to construct pairwise miRNA-miRNA and disease-disease enrichment networks, and identified 23 central miRNAs that could underlie major regulatory functions in the human genome. miR2Trait is available as an open-source command-line interface in Python3 (URL: https://github.com/miR2Trait) with a companion wiki documenting the scripts and data resources developed, under MIT license for commercial and non-commercial use. A minimal web-based implementation has been made available at https://sas.sastra.edu/pymir18. Supplementary information is available at: https://doi.org/10.6084/m9.figshare.8288825.v3.

## INTRODUCTION

MicroRNAs (miRNAs) are key elements of post-transcriptional regulation in the genomic architecture of both prokaryotes and eukaryotes. They are short non-coding RNAs of about 18–25 nucleotides, first observed in *C. elegans* (*Lee, Feinbaum & Ambros, 1993*). In their canonical role in RNA silencing, the miRNA binds to the cognate mRNA and destabilizes it, thereby priming the transcript for degradation. A single miRNA is capable of silencing the expression of several genes by relaxing the specificity of hybridisation. MiRNA-based regulation plays crucial roles in health, and miRNA dysregulation is a common mechanism in the etiology of complex diseases, viz. cardio-vascular, auto-immune and neuro-degenerative diseases, and cancers (*Min et al., 2009*; *Calin et al., 2004*; *Tai, 2011*). About half of all miRNA genes are present in cancer-related genetic loci (*Calin et al., 2004*),

Corresponding author
Ashok Palaniappan,
apalania@scbt.sastra.edu

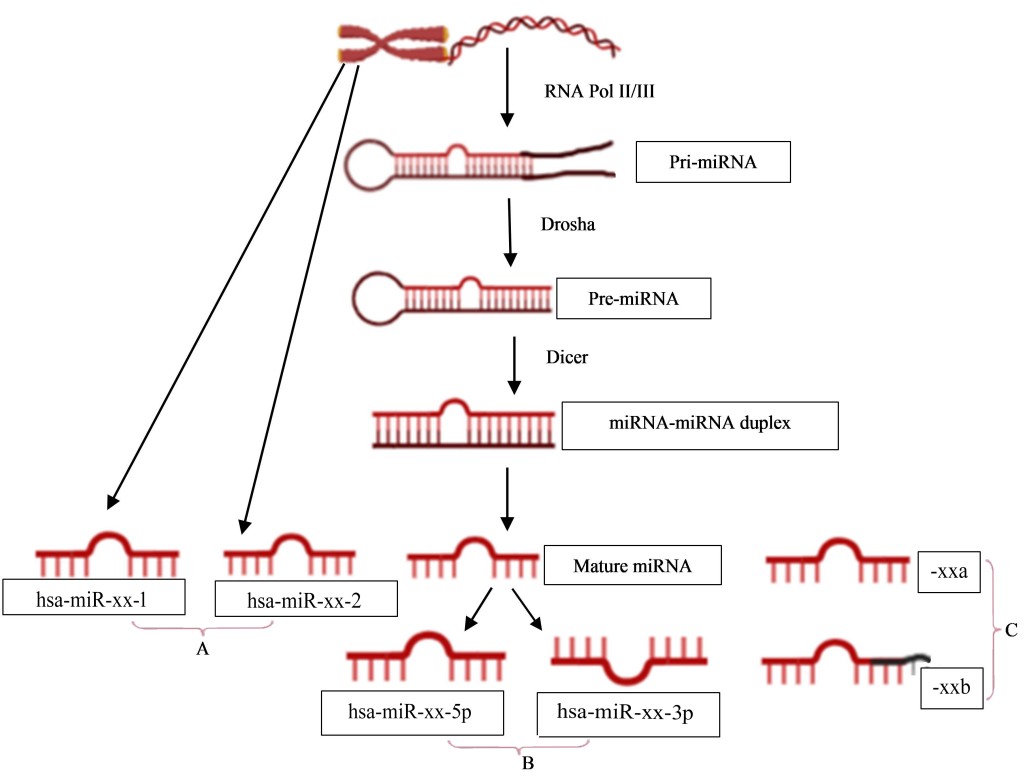

**Figure 1 miRNA nomenclature mirrors the pathway of its biogenesis.** (A) identical miRNAs encoded by different genetic loci (1,2, …); (B) source arm of the miRNA duplex (5p or 3p); and (C) nearly identical miRNAs that differ in one or two positions (a,b, …).

and they are known to play key roles in tumorigenesis and cancer progression (*Liu et al., 2018*; *Hayes, Peruzzi & Lawler, 2014*; *MacFarlane & Murphy, 2010*).

The naming of human miRNAs in the miRBase registry (https://www.mirbase.org) is related to the miRNA maturation pathway (Fig. 1). In brief, miRNA subtyping with numbers (−1, 2, …) indicates identical miRNAs that derive from different genomic regions; miRNA subtyping with letters (a, b, …) indicates nearly identical miRNAs that differ in only one or two positions; and miRNA designations -5p and -3p refer to the source arm of the miRNA duplex from which the mature miRNA is derived. It is the mature miRNA that binds to the mRNA and inhibits/regulates protein translation (*MacFarlane & Murphy, 2010*). It is noteworthy that the sequence responsible for miRNA-specificity of the mRNA is not translated, thus separating the protein-coding region of a gene from the miRNA-binding site in sequence space.

Existing databases of miRNA–disease associations involve tedious manual curation or are limited in scope and evolution. miR2Disease was one of the earliest catalogues of miRNA-disease associations, providing relationships between 349 miRNAs and 163 diseases (*Jiang et al., 2009*). HMDD was developed by *Li et al. (2014)* to provide a more comprehensive database of miRNA-disease associations. It used text mining followed by manual annotation for identifying miRNA-disease connections. The current version of
HMDD, version 3.2, contained associations between 1206 miRNAs and 893 diseases, and included miRNA subtype information (*Huang et al, 2019*). Only one arm of the mature miRNA, indicating the -5p or -3p source, is predominant *in vivo*, a vital annotation that is missing in the HMDD databases. In this work, we have addressed the limitations of previous efforts, and designed an algorithm to uncover the spectrum of miRNA–disease associations. Our approach is based on integrating two expert-curated databases on a common key. The resulting resource, miR2Trait, is comprehensive and representative, uncovers novel relations, and provides assorted tools for investigating miRNA-disease associations. miR2Trait can be queried by both miRNA and disease, and is available as both a web server (https://sas.sastra.edu/pymir18) and command-line resource (https://github.com/miR2Trait/ or http://doi.org/10.5281/zenodo.7002878).

## MATERIALS & METHODS

### miRNA: disease mapping

Two expert curated databases were used as the source databases in the creation of miR2Trait: (1) miRTarbase, a database of experimentally validated miRNA–target interactions (MTIs), with >13,400 entries (*Chou et al., 2018*); and (2) DisGeNET, a widely used and standardized knowledge management platform with >1,000,000 gene–disease associations (*Piñero et al., 2015*). Both databases involved the mapping of genetic information, in one case to miRNAs, and in the other, to diseases. This provided a clue to relating miRNAs and disease via the bridge of genetic information (Fig. 2). In-house Python scripts were used to extract the gene:miRNA and the gene:disease mappings from the respective databases as dictionary data structures. The two dictionaries were merged on 'gene' to establish a dictionary of miRNA:disease mappings. Similarly, an inverse dictionary of disease:miRNA mappings was created.

### Database creation

The dictionaries obtained above were used to populate relational tables in the construction of two databases: (i) miRNAs and their associated diseases, and (ii) diseases and their associated miRNAs. MySQL was used for encoding the databases.

### Spectrum width calculation

The number of diseases with which an miRNA is associated is indicative of the breadth of its regulatory impact, which could flag master regulator miRNAs (or regulatory hubs). The disease spectrum width (DSW) of the ith miRNA is calculated after *Qiu, Chen & Cui (2012)*:

$$\text{DSW}(i) = d_\text{i}/D_\text{N} \tag{1}$$

where $d_\text{i}$ is the number of diseases associated with miRNA i and $D_\text{N}$ is the total number of diseases in the database. Conversely the number of miRNAs associated with a given disease could be indicative of a complex multifactorial pathology, and the miRNA spectrum width (MSW) of the jth disease is given by:

$$\text{MSW}(j) = m_\text{j}/m_\text{N} \tag{2}$$

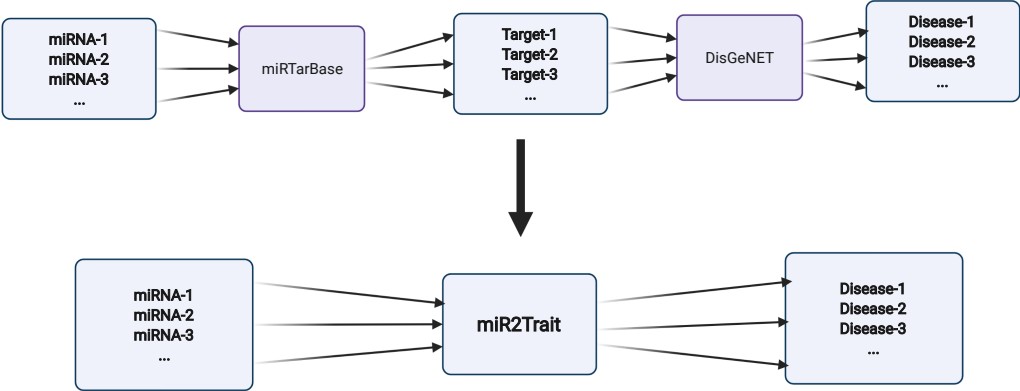

**Figure 2** **Design of miR2Trait.** A back-to-back double bipartite network could be reduced to a single bipartite graph by reductive mapping on the common key. Here, the target nodes of one database serve as the source nodes of the second database. By bridging on these 'genes', a mapping between the source nodes of the first database (miRNAs) and the target nodes of the second database (diseases) could be obtained. Created with BioRender.com.

where $m_j$ is the number of miRNAs associated with the jth disease and $M_N$ isthe total number of miRNAs in the database.

## miRNA list enrichment analysis

Given a list of miRNAs that are collectively dys-regulated, it would be of interest to identify which diseases would be enriched (or more likely to occur). Statistical techniques to quantify such enrichment for genes as well as miRNAs exist (*Li et al., 2018*; *Backes et al., 2016*; *Çorapçıoğlu & Oğul, 2015*; *Rivals et al., 2007*). Here we implement an open-source tool to identify diseases in the miR2Trait database that are enriched for an input set of miRNAs. The method uses a hypergeometric test to quantify the enrichment of each disease in miR2Trait for the given set of miRNAs (in this case identical to the Fisher's exact test) (*Fisher, 1922*). The test follows the construction of the Fisher contingency table for each disease, as illustrated in Table 1.

The statistical significance (*p*-value) of enrichment is computed using Python scipy routines (http://www.scipy.org/), and diseases that pass a user-specified adjusted *p*-value threshold (with an option to choose among three methods for multiple hypothesis correction) are returned sorted by significance. In addition to significance, both the web server and standalone tool calculate the effect size (log-fold) of each enrichment.

## Diseasome analysis

Given an input set of diseases related in some way (called a diseasome), it would be of interest to find shared dysregulated miRNAs. We approached this problem with an abundance analysis of miRNAs in the diseasome, and identifying miRNAs that passed a user-specified count. Such a set of miRNAs could constitute master regulators acting on common pathways in the specified set of diseases. If we are interested in the miRNA spectra of umbrella disease terms such as 'carcinoma', instead of a specific set of diseases, this is also allowed. In this case, diseases containing the general keyword are first identified,

**Table 1  Fisher's contingency table for miRNA list enrichment analysis for one disease in the database.** Each cell value represents the count for the intersecting row and column.

| miR2Trait | miRNAs associated with disease | miRNAs not associated with disease | (Sum) |
|---|---|---|---|
| miRNAs in input | a | b | a+b |
| miRNAs not in input | c | d | c+d |
| (Sum) | a+c | b+d | n |

and then the miRNA abundance for this diseasome is computed in a manner similar to above. A CLI script to compute the miRNA profile of a specified diseasome is provided with miR2Trait (https://github.com/miR2Trait).

**miR2Trait web server and standalone tools**

A web interface using PHP to connect the user HTML front-end with the MySQL backend was created to offer access to all the services developed, including the two databases and the assorted tools. MiRNA list enrichment analysis (miLEA) was implemented using the python DBI module, and receives the user's miRNA list from the HTML form via PHP intermediate. The command-line version of the enrichment analysis provides adjusted *p*-value calculation using Bonferroni, Holm-Bonferroni or Benjamini–Hochberg (default) correction. Other CLI scripts developed include:

(i) Query standalone miR2Trait by miRNA or disease

(ii) Retrieve miRNA-specific information from miRBase by user-given miRNA name(s) or sequence(s).

(iii) Network creation: creating an miRNA-miRNA/disease-disease adjacency network from miR2Trait dictionaries

(iv) Postprocessing constructed networks for consensus central nodes (miRNAs or diseases)

The user is referred to the wiki (https://github.com/miR2Trait/miR2Trait/wiki) for reproducible examples and use-cases of all the scripts developed.

## RESULTS

The mapping sizes extracted from the respective databases is given in Table 2. In summary, 2,599 miRNAs were mapped to 15,062 genes using miRTarbase, and 8,819 genes were mapped to 13,075 diseases using DisGeNET. Merging the two relations yielded a mapping between 2,595 miRNAs and 11,689 diseases (Table 2). This mapping laid the foundation for the miRNA:disease and disease:miRNA databases, which could be freely downloaded as csv files from aforementioned URLs. The density function of the mappings is shown in Fig. 3.

Using the miRNA:disease db, miRNAs with the top ten DSWs were identified (Table 3). Such miRNAs are potential master (or hub) regulators at the intersection of multiple pathways, indicatively contributing to one-fifth of all diseases. Using the disease:miRNA db, diseases with the top ten MSWs were identified (Table 4). Complex neurological

**Table 2  Establishing the miR2Trait mappings.** Only four miRNAs from miRTarBase did not map to any disease in the final analysis.

| Database/resource | Mapping relation | Mapping size |
|---|---|---|
| miRTarBase | miRNAs →Target genes | 2,599 → 15,062 |
| DisGeNET | Genes → Diseases | 8,819 → 13,075 |
| miR2Trait | miRNAs → Diseases | 2,595 → 11,689 |

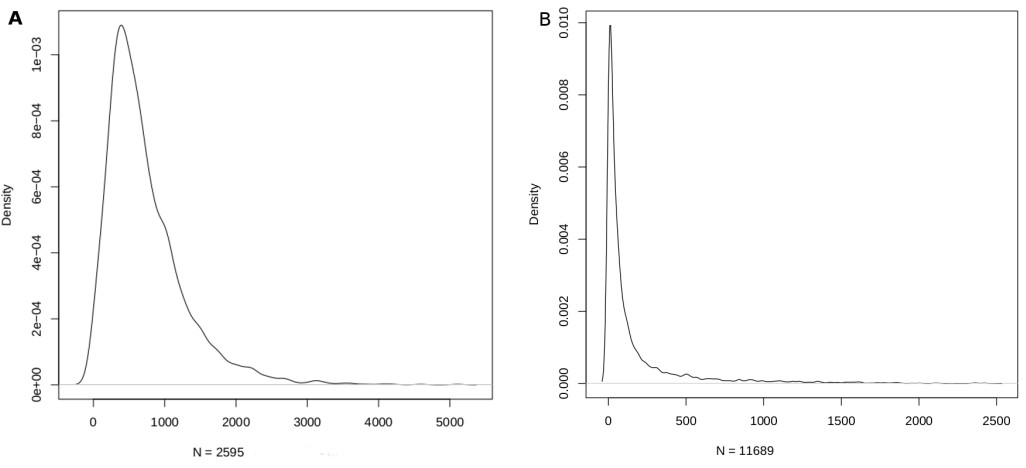

**Figure 3  Density function plots of (A) miRNA-disease; and (B) disease-miRNA mappings.** The mode of the number of diseases/miRNA is about 389 and the mode of the number of miRNAs/disease is about 9.

**Table 3  DSW of miRNAs.** miRNA corresponding to the top ten DSWs are shown.

| S.No | miRNA | DSW | No. of associated diseases |
|---|---|---|---|
| 1 | hsa-miR-335-5p | 0.22 | 2,530 |
| 2 | hsa-miR-26b-5p | 0.20 | 2,321 |
| 3 | hsa-miR-124-3p | 0.19 | 2,175 |
| 4 | hsa-miR-16-5p | 0.18 | 2,110 |
| 5 | hsa-miR-92a-3p | 0.17 | 1,974 |
| 6 | hsa-miR-1-3p | 0.16 | 1,913 |
| 7 | hsa-miR-17-5p | 0.16 | 1,912 |
| 8 | hsa-let-7b-5p | 0.16 | 1,864 |
| 9 | hsa-miR-155-5p | 0.16 | 1,856 |
| 10 | hsa-miR-93-5p | 0.16 | 1,766 |

conditions sweep >90% of all miRNAs, reflecting multifactorial, far from understood aetiologies.

The miRNA Enrichment Analyzer computes both the odds-ratio size and significance *p*-value for a given set of miRNAs. The input could be variable in the number of miRNAs provided. We used "hsa-miR-346 hsa-miR-26a-5p hsa-miR-7-5p hsa-miR-34a-5p" as

**Table 4  MSW of diseases.** Diseases corresponding to the top ten MSWs are shown.

| S.No | Disease | MSW | No. of associated miRNAs |
|------|---------|-----|--------------------------|
| 1 | Autosomal recessive predisposition | 0.97 | 2,491 |
| 2 | Schizophrenia | 0.96 | 2,466 |
| 3 | Intellectual disability | 0.95 | 2,452 |
| 4 | Low intelligence | 0.94 | 2,421 |
| 5 | Dull intelligence | 0.94 | 2,421 |
| 6 | Poor school performance | 0.94 | 2,421 |
| 7 | Mental deficiency | 0.94 | 2,421 |
| 8 | Mental retardation | 0.94 | 2,421 |
| 9 | Mental and motor retardation | 0.92 | 2,381 |
| 10 | Global developmental delay | 0.92 | 2,381 |

**Table 5  miRNA list enrichment analysis.** Results for the query: ''hsa-miR-2355-3p hsa-miR-214-3p hsa-miR-4801'', and containing the word 'neoplasm' are shown, sorted by adjusted significance ($q$-value). An odds ratio (effect size) of 'Inf' indicates that the entire set of miRNAs is contained in the disease definition.

| Disease | q-value | Odds-ratio |
|---------|---------|------------|
| Colorectal Neoplasms | 0.014 | Inf |
| Neoplasm of the anterior pituitary | 0.020 | 74.50 |
| Stomach Neoplasms | 0.022 | Inf |
| Prostatic Neoplasms | 0.027 | Inf |
| Malignant neoplasm of ovary | 0.034 | 43.32 |
| Biliary Tract Neoplasm | 0.035 | 42.83 |

a query with an adj. $p$-value (*i.e.*, $q$-value) cutoff of 0.05. This returned a set of 4,121 significantly enriched diseases. As another example, we used a set of three random miRNAs: ''hsa-miR-2355-3p hsa-miR-214-3p hsa-miR-4801'' as a query. This returned 830 significantly enriched diseases (Table 5). Dysregulation of key cancer-associated miRNAs (*Chan, Prado & Weidhaas, 2011*) could contribute to the emergence of cancer hallmarks (*Hanahan & Weinberg, 2011*) and could be tested likewise.

Analysis of diseasomes for over-represented miRNAs could yield valuable hypotheses for further research, in terms of both pathophysiology and therapeutic options. In this line, we pursued the investigation of diseasomes related to cancer. Since the evidence for miRNA involvement in all stages of cancer including tumorigenesis, progression and metastasis is well-established, we probed miR2Trait for the keyword 'Neoplasm' diseasome, and performed an miRNA occurrence analysis. We also performed occurrence analysis for diseasomes identified using the keywords 'Carcinoma', 'Sarcoma', 'Lymphoma', and 'Leukemia'. Table 6 shows the results of these analyses for 25 miRNAs ranked by the 'Neoplasm' occurrence analysis. MiRNAs documented in the literature as playing key roles in various cancers appeared in the top miRNAs for each keyword (*Sandoval-Bórquez et al., 2017*; *Miyamoto et al., 2016*; *Zhang et al., 2015*; *Wang et al., 2018*; *Cloonan et al., 2008*; *Xu et al., 2014*; *Robertson et al., 2014*; *Fabbri et al., 2016*; *Garzon et al., 2008*; *Marcucci et al.,*

**Table 6 Diseasome analysis of miRNA over-abundance.** Statistics for only the top 25 miRNAs from 'Neoplasm' occurrence analysis are shown with respect to cancer-related keyword searches.

| 'Neoplasm' rank | miRNA | Carcinoma | Leukemia | Lymphoma | Sarcoma |
|---|---|---|---|---|---|
| 1 | hsa-miR-34a-5p | 43 | 21 | 17 | 16 |
| 2 | hsa-miR-16-5p | 47 | 24 | 15 | 18 |
| 3 | hsa-miR-124-3p | 42 | 16 | 18 | 21 |
| 4 | hsa-miR-335-5p | 41 | 25 | 17 | 20 |
| 5 | hsa-miR-21-5p | 45 | 12 | 17 | 15 |
| 6 | hsa-miR-155-5p | 48 | 17 | 17 | 18 |
| 7 | hsa-miR-19a-3p | 38 | 16 | 15 | 13 |
| 8 | hsa-miR-17-5p | 47 | 24 | 16 | 16 |
| 9 | hsa-miR-125b-5p | 36 | 16 | 15 | 21 |
| 10 | hsa-let-7b-5p | 35 | 15 | 13 | 20 |
| 11 | hsa-miR-221-3p | 35 | 15 | – | 17 |
| 12 | hsa-miR-15a-5p | 40 | 16 | 13 | 16 |
| 13 | hsa-miR-130b-3p | 33 | 11 | 11 | 11 |
| 14 | hsa-miR-26b-5p | 39 | 15 | 14 | 18 |
| 15 | hsa-miR-181a-5p | 37 | 17 | 14 | 16 |
| 16 | hsa-miR-19b-3p | 36 | 13 | 14 | 13 |
| 17 | hsa-miR-106a-5p | 42 | 13 | 17 | 15 |
| 18 | hsa-miR-192-5p | 41 | 20 | 11 | 14 |
| 19 | hsa-miR-106b-5p | 43 | 15 | 16 | 11 |
| 20 | hsa-miR-218-5p | 37 | 26 | 15 | 16 |
| 21 | hsa-miR-24-3p | 36 | 14 | 14 | 12 |
| 22 | hsa-miR-27a-3p | 41 | 21 | 12 | 10 |
| 23 | hsa-miR-20a-5p | 43 | 20 | 16 | 16 |
| 24 | hsa-miR-454-3p | 32 | 10 | 10 | 10 |
| 25 | hsa-miR-30a-5p | 47 | 16 | 13 | 14 |

*2008*; *Calin et al., 2005*; *Stefano et al., 2010*; *Jima et al., 2010*). Databases devoted to miRNA associations in cancer have been developed (*Yang et al., 2017*; *Sarver et al., 2018*; *Ahmed et al., 2018*), and our work here would augment efforts in this direction. Expression of specific microRNAs has been recognized as a vital diagnostic/prognostic biomarker and therapeutic target in a variety of cancers (*Blenkiron et al., 2007*; *Sempere et al., 2007*; *Michael et al., 2006*; *Iorio et al., 2005*; *Lowery et al., 2009*; *Lebanony et al., 2009*; *Ueda, 2010*; *Fridman et al., 2010*; *Gutiérrez et al., 2010*).

## DISCUSSION

### Benchmarking

Dedicated miRNA omics databases have been developed to aid researchers unravel the role of miRNAs in biological processes (for *e.g.*, see *Ruepp et al., 2010*; *Keller et al., 2011*). Computational methods have also been advanced for inferring miRNA-disease connections (*Chen, Liu & Yan, 2012*; *Xuan et al., 2013*; *Qu et al., 2018*). HMDDv3.2 is a
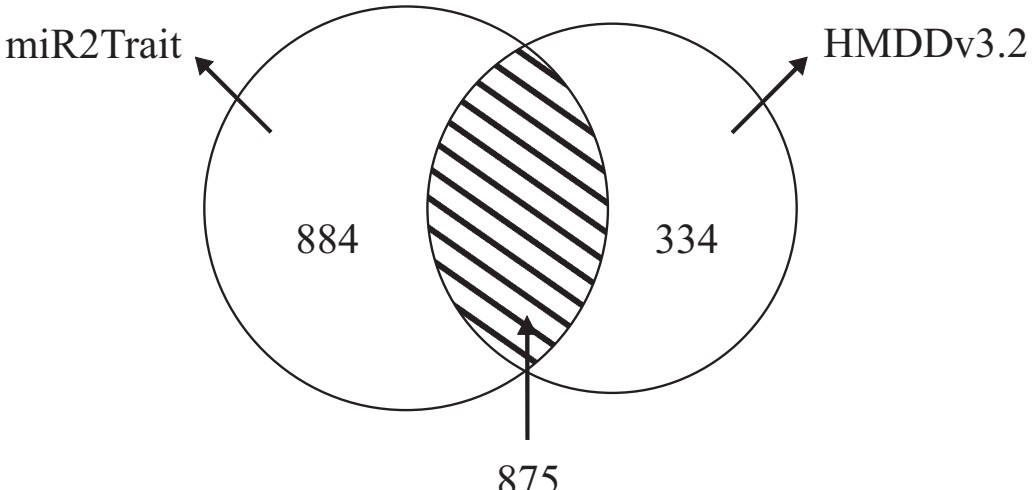

**Figure 4** **Benchmarking against HMDDv3.2.** The miRNA content in both miR2Trait and HMDDv3.2 have been compared here, and a Venn diagram shown with the unique complements and the overlap between the two databases. A substantial number of miRNAs, 875 in total, are common to both the databases. In such cases, miR2Trait could provide complementary information and analysis of the miRNAs of interest. miRNAs unique to miR2Trait outnumber the miRNAs unique to HMDDv3.2 by a factor of 2.5.

manually curated database of miRNA-disease associations based on text-mining. While miR2Trait contains the full mature miRNA information, HMDD does not provide subtype information such as -5p or -3p; 1a or 1b. The miRNA arm designation (5p or 3p) is a major ingredient of disease etiology, and key to naming miRNAs (for *e.g.*, the ones with the top DSWs in Table 3). To compare the databases, it is therefore necessary to 'normalize' the miRNA nomenclature between the databases (by suppressing the subtype information in miR2Trait). This contracted the unique miRNAs in miR2Trait to 1,759 in number. Following this, the databases were compared, and the results are shown in Fig. 4. Even after contraction, miR2Trait appears more comprehensive in the catalog of miRNA-disease associations. miR2Trait-unique miRNAs are 2.5-fold more in number than HMDDv3.2 - unique miRNAs (884 vs 334). The unique miRNA complements suggest directions for research. Some of the diseases missed by miR2Trait could be those that involve deleterious mutations in the miRNA-binding region of the gene itself. Finally, the significant size of the overlap between the two databases increases confidence in the methods advanced here, which are also easily extensible with updates to the underlying primary databases. These observations further suggest the use of miR2Trait to complement HMDD.

## Network analysis

The availability of two-way mappings between miRNAs and diseases with various analysis tools paves the way to varied investigations. One direction would be to identify pairs of miRNAs that lead to clusters of similar disease phenotypes. We addressed this question by using the miRNA enrichment analyzer discussed herein to compute the disease enrichment scores (2-tuples of BH-adjusted $p$-value and odds-ratio) of each miRNA-miRNA pair

using Fisher's exact test. This process is iterated over the entire set of miRNA-miRNA pairs (2595*2595) to obtain an adjacency matrix of ~3 million interactions. This matrix was used to construct the corresponding network made of significant edges weighted by the odds-ratio. The interactions were sorted by $-\log_{10}$ $p$-value and the top 10,000 interactions were used to construct the miRNA-miRNA adjacency network, yielding 2,442 nodes (and 10,000 edges). The network was read into Cytoscape (*Shannon et al., 2003*), and centrality analysis was performed using CytoNCA (*Tang et al., 2015*). The weighted versions of the following seven centrality measures were used: betweenness, closeness, eigenvector, degree, network, information, and local average centralities. The top 100 miRNAs from each centrality measure were used to obtain the consensus (five or more measures) top miRNAs (Table 7; algorithm details and Python script given here: https://github.com/miR2Trait/miR2Trait/wiki/Top-Nodes-from-Adjacency-Networks). Edge-percolation centrality of the network was also computed using cytoHubba to rank the miRNAs (*Chin et al., 2014*). The top 23 miRNAs contained 14 consensus miRNAs and the top 75 contained all but one of the consensus miRNAs. Such a consensus list could identify the critical regulatory miRNAs in human for further investigation. A disease-disease adjacency network was similarly constructed, and both the above adjacency networks and the scripts to perform the network construction and analysis are available from miR2Trait (https://github.com/miR2Trait).

## Deployment

The miR2Trait web server and standalone CLI could be queried by either disease or miRNA. The standard use-case would be to identify miRNAs associated with some disease. For instance, a search by disease for 'Myocardial infarction' returned 1,445 associated miRNAs. The top hits included hsa-miR-302a-3p, hsa-miR-302b-3p, hsa-miR-302c-3p, hsa-miR-302d-3p, hsa-miR-367-3p and hsa-miR-367-5p, all of which are well-documented in the literature (*Sun et al., 2017*). The top hits for a search by disease of 'Alzheimer's disease' included hsa-miR-9, hsa-miR-29a, hsa-miR34a, hsa-miR-106b, hsa-miR-125b, hsa-miR-146a, and hsa-miR-155, again all of which are well-documented in the literature (*Leidinger et al., 2013*). As a last example, the top hits for a search by disease of 'Diabetes mellitus' included hsa-miR-577, hsa-miR-37a, hsa-miR-375, hsa-miR-181a, hsa-miR-17, and hsa-miR-24, all of which are reported in the literature (*Kim & Zhang, 2019*; *Guay et al., 2011*). The search term is case-insensitive, matches anywhere, and interpreted according to regex grammar (https://dev.mysql.com/doc/refman/8.0/en/regexp.html#operator_regexp). Such searches might yield miRNAs lacking documented disease connections, and offer hypotheses for future investigations. In this regard, miR2Trait provides a script to retrieve known information about miRNAs from miRBase upon querying with the miRNA name or sequence (https://github.com/miR2Trait/miR2Trait/wiki/Querying-miRNA-information-from-miRBase).

In summary, we have developed miR2Trait, a comprehensive resource for investigating miRNA-disease associations, based on a simple yet surprisingly effective technique for fusing two curated primary databases. One outcome of the approach has been the enhanced

**Table 7  Consensus identification of key miRNAs in disease development.** Seven centrality measures were used to identify the top 100 nodes in the miRNA-miRNA adjacency network and their consensus is determined based on the number of measures that agree upon an miRNA (agreement ≥ 5). The consensus miRNAs appear central even by the Edge-percolation centrality (EPC) measure.

| SNo | miRNA | Agreement | EPC rank, score |
|---|---|---|---|
| 1 | hsa-miR-1253 | 7 | 36, 208 |
| 2 | hsa-miR-6755-5p | 7 | 13, 247 |
| 3 | hsa-miR-3911 | 6 | 62, 188 |
| 4 | hsa-miR-6804-5p | 6 | 16, 232 |
| 5 | hsa-miR-4689 | 6 | 12, 247 |
| 6 | hsa-miR-873-5p | 6 | 11, 247 |
| 7 | hsa-miR-548aq-3p | 5 | 8, 254 |
| 8 | hsa-miR-548x-3p | 5 | 6, 257 |
| 9 | hsa-miR-548ah-3p | 5 | 3, 260 |
| 10 | hsa-miR-6885-3p | 5 | 28, 222 |
| 11 | hsa-miR-548aj-3p | 5 | 7, 255 |
| 12 | hsa-miR-548j-3p | 5 | 5, 258 |
| 13 | hsa-miR-548ae-3p | 5 | 2, 261 |
| 14 | hsa-miR-543 | 5 | 41, 203 |
| 15 | hsa-miR-194-5p | 5 | 20, 229 |
| 16 | hsa-miR-3140-3p | 5 | 73, 181 |
| 17 | hsa-miR-3156-3p | 5 | 43, 202 |
| 18 | hsa-miR-548am-3p | 5 | 4, 260 |
| 19 | hsa-miR-892b | 5 | 10, 253 |
| 20 | hsa-miR-4280 | 5 | n/a |
| 21 | hsa-miR-802 | 5 | 25, 224 |
| 22 | hsa-miR-877-5p | 5 | 1, 266 |
| 23 | hsa-miR-425-3p | 5 | 26, 223 |

interpretability and transparency of the resource constituents, aspects that are lost in more complex methods to arrive at reliable miRNA-disease associations.

## CONCLUSIONS

MicroRNAs play key roles in health and the development of disease by virtue of their crucial regulatory activities. To document miRNA-disease associations, we used a novel method integrating two curated and validated sources of miRNA-gene and gene-disease relations, and created vastly expanded databases of miRNA-disease and disease-miRNA associations. A set of tools to interrogate these databases and uncover novel findings has been developed. Taken together, miR2Trait provides a starting point for the investigation of the miRNA-ome in the context of disease. A web-server interface to all the functionalities has been developed and is available at https://sas.sastra.edu/pymir18/. The service allows for flexible querying using regular expression-based pattern matching. It includes a tool for miRNA-enrichment analysis that would enable the identification of statistically over-represented diseases in a user-given list of miRNAs. This would help formulate hypotheses of miRNA-mediated

common mechanisms underlying multiple disease dysregulation pathologies. A number of examples have been discussed to illustrate the use-cases for the resource. The resource has also been benchmarked with the best alternative miRNA-disease database (HMDDv3.2). Further, the source code for the entire project is made freely available, and includes a plethora of tools for working with the resource, along with an extensive wiki. All the data developed in the project are freely available as supplementary information, including the miRNA-miRNA and disease-disease adjacency networks.

## ACKNOWLEDGEMENTS

We would like to thank the School of Chemical & Biotechnology, SASTRA University for infrastructure and computing support.

### Funding

This work was supported by the Science and Engineering Research Board, DST India (no. EMR/2017/000470). The funders had no role in study design, data collection and analysis, decision to publish, or preparation of the manuscript.

### Grant Disclosures

The following grant information was disclosed by the authors:
Science and Engineering Research Board, DST India: EMR/2017/000470.

### Competing Interests

The authors declare there are no competing interests.

### Author Contributions

- Poornima Babu performed the experiments, analyzed the data, prepared figures and/or tables, authored or reviewed drafts of the article, and approved the final draft.
- Ashok Palaniappan conceived and designed the experiments, performed the experiments, analyzed the data, prepared figures and/or tables, authored or reviewed drafts of the article, and approved the final draft.

### Data Availability

The code is available at GitHub: https://github.com/miR2Trait.
The data is available at FigShare: Babu, Poornima; Palaniappan, Ashok (2021): miR2Trait: an integrated resource for mining miRNA - disease associations. figshare. Online resource. https://doi.org/10.6084/m9.figshare.8288825.v3.

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
