# Peer review of "miR2Trait: an integrated resource for investigating miRNA-disease associations"

_PeerJ, doi:10.7717/peerj.14146_

## Round 0.1 · original submission · Major Revisions

Two experts assessed your manuscript and found relevant the bioinformatics resource but also raised some concerns that should be addressed before acceptance for publication. In particular, the accessibility to scripts and tools, and the improvement of the interface should be addressed in a revised version of the manuscript.

·

Basic reporting

The work by Babu and Palaniappan presents the generation of a resource for the investigation of the association between miRNA and disease. The rationale for designing this database is to establish a catalogue of miRNAs and diseases using two curated databases.
The resources are available online and accessible. The manuscript is well written.
In the following sections, this reviewer highlights some concerns that authors are encouraged to address.

Experimental design

The generation of a database is a complex endeavor. However, in this case, the database has issues that this reviewer finds important and reduces the accessibility and the value of the output derived of the user query. Also, the output is limited to a long list of diseases, but no more information is provided, even the total of the hits found. Please find these comments in the next section.

Validity of the findings

Respectfully, this reviewer finds the interface of the database not so friendly and no difference between “query by miRNA” and “miRNA enrichment analysis” for highly characterized miRNA such as let-7. Here, the expression of key target genes or the expression of the query miRNA would be of value. The list of diseases associated with the miRNA is complicated to grasp as a functional database, especially that no more information is provided or any other link between data can be formulated. This reviewer thinks that perhaps the problem is also the way is presented in the manuscript, I think that authors can generate a more comprehensive guide of the data presented in the database as well as the output. Following this line of thought, the mir2disease database have a more comprehensive query and result analysis (even though is more limited in other aspects, in good agreement with the authors), for example, Let-7 provides the different isoforms and when analyzing there is the diseases associated with this miRNA and the references related to this miRNA. The database presented in this report even though is robust regarding the contained data, the accessibility is a serious issue and obtaining a very long list of the diseases associated with each derived miRNA for further experimental validation per disease is limited, perhaps having the top ten novel miRNAs is more useful rather than querying each “miRNA of interest”.
Figure 3 is of little value as it is. This reviewer thinks that showing the tools in more detail is more useful, rather than only the construction of the user interface. Also, this reviewer suggests fusing figure 2 and 3.
Figure 5 legend requires a revision, this reviewer got lost with it.

Additional comments

In line 33 I recommend using small or short instead of diminutive, this implies that are extremely small and there are miRNAs that are unusually long.
In line 76-77 it is unclear how the merge was done, this reviewer regrets to not understanding the meaning of ‘gene’ here, I suggest making the line “…were merged to each gene to establish a two sets of dictionary-based query search” if I understand this correctly. Also, “vein” is not correct here, “avenue” perhaps is more suitable.
In lines 84 and 89, define DSW and MSW. Peer J is a broad readership journal, so all concepts must be correctly defined.
Line 253, italics are missing in C. elegans.

Reviewer 2 ·

Basic reporting

The authors of the manuscript present an implementation of a database that links microRNAs to candidate diseases. They accomplish this by linking to carefully expert-curated databases: miRTarbase, a database of experimentally validated miRNA-target interactions, and DisGeNET, a management platform with gene-disease associations. miRTarbase links known miRNA to specific genes while DisGeNET contains information on the link between genes and diseases. Through their implementation, the authors connect both databases by means of the common knowledge (genes) thus associating knowledge of miRNA to diseases.
The paper is concise and well written in clear and unambiguous professional English, with the appropriate background to put in context their database implementation. The literature references are, to the best of my knowledge, sufficient and cover the subject matter.
My only concerns are about the availability of tools and data (see experimental design).

Experimental design

The database creation and statistical analysis of the generated networks, to the best of my knowledge, look correct.
My only concerns are regarding the availability of all the tools and data, as well as missing features that would allow for better usability.
-All the scripts and tools should be available at their GitHub site and not "available from the authors upon request".
-The command-line resources at GitHub should include a more detailed manual or wiki. In its current state, it only lists the scripts with a very basic description of each one.

A feature or functionality that, in my opinion, is missing, is the retrieval of the corresponding sequence of the miRNA. As it is, the database can retrieve miRNAs names from diseases or vice-versa (diseases associated to a query miRNA). In both directions, the reader only obtains (or starts from) the name of the miRNA. It would be useful to link these names to the actual sequences of the corresponding miRNA or even their secondary structure, as shown in databases such as miRBase (https://www.mirbase.org/). The possibility of searching by sequence instead of just by miRNA name would be also useful [These are suggestions and I am aware that its implementation may be outside the scope of the presented tool or even too complicated to achieve so I understand if these are not included on a revised version].

As a note, I would remove lines 210-226 and instead point the reader to resources with regex explanations or tutorials. In its current state, these lines present the reader with complex searches but with too brief and too ambiguous explanations of their use.

Validity of the findings

As far as I am aware, the tools presented seem to achieve the intended goals and the statistical analysis performed are appropriate and valid.

---

## Round 0.2 · accepted · Accept

The authors addressed all the Reviewers' comments and the manuscript is now suitable for publication in this journal.

·

Basic reporting

After the revision, authors have addressed most of my concerns. The manuscript has been improved and now has more clarity.

Experimental design

The data base has been improved and the inclusion of a Wiki has facilitated the use of the code. During the revision, the tests this reviewer did, the software was functioning fine. The software was tested again and works fine. I only suggest expanding the WIki a bit more for less experienced users.

Validity of the findings

The database has many potential expansions that may be more powerful overtime. I encourage authors to keep building the database further and perhaps with collaborations, expression/abundance can be incorporated to have a more robust tool.

Additional comments

No additional comments.